# The Impact of Omega-3 Supplements on Non-Surgical Periodontal Therapy: A Systematic Review

**DOI:** 10.3390/nu14091838

**Published:** 2022-04-28

**Authors:** Luísa Martins Miller, Flávia Benetti Piccinin, Ubele van der Velden, Sabrina Carvalho Gomes

**Affiliations:** 1Post-Graduate Program, School of Dentistry, Federal University of Rio Grande do Sul, Porto Alegre 90035-003, Brazil; luisamartinsmiller@hotmail.com; 2Department of Periodontology, School of Dentistry, Institute of Higher Education of Santo Ângelo (IESA), Santo Ângelo 98801-015, Brazil; flapiccinin@gmail.com; 3Department of Periodontology, Academic Centre for Dentistry Amsterdam, University of Amsterdam, 1081 LA Amsterdam, The Netherlands; u.vd.velden@acta.nl; 4Conservative Dentistry Department, School of Dentistry, Federal University of Rio Grande do Sul, Porto Alegre 90035-003, Brazil

**Keywords:** fatty acids, omega 3, periodontal diseases, periodontitis

## Abstract

Aim: This systematic review examined the additional effect of taking omega-3 supplements on periodontal therapy. Methods: The focused question was “What is the possible effect of omega-3 supplementation concomitant to non-surgical periodontal therapy on clinical periodontal parameters?” Databases Cochrane, Embase, Google Scholar, PubMed, and Web of Science (January-July 2021) were searched to identify appropriate studies. Randomized clinical trials (RCT) about non-surgical therapy with omega-3 supplementation, with at least 3 months of supplementation period were included. Cochrane risk of bias tool version 2 and Grading of Recommendations Assessment, Development, and Evaluation were used. Results: A total of 1556 studies were found, of which eight studies met the inclusion criteria. All eight studies evaluated periodontal probing depth and clinical attachment loss; plaque and gingival inflammation were evaluated in seven studies. High variety of omega-3 dosage, different study lengths, questionable results from periodontal therapy (including test and control groups), high risk of bias and moderate quality of evidence prevented a satisfactory conclusion regarding the benefits of omega-3 supplementation. The studies’ high heterogeneity avoided meta-analysis. Conclusion: Notwithstanding all limitations, the promising effects of omega-3 supplementation presented in two six-month studies encourage performing RCT with better-defined treatment protocols and greater methodological rigor.

## 1. Introduction

The oral cavity harbors numerous and distinct microorganisms, collectively referred to as the oral microbiome. This microbiota is found in various oral niches, such as the tongue, saliva, mucosa, and teeth [1], and typically live in harmony and balance with the host [2]. The behavior of the biofilm on teeth is influenced by factors such as lifestyle changes, antimicrobial therapies, and the inflammatory status of the host [2]. This condition may lead to irreversible destruction of periodontal support tissues, resulting in the establishment of periodontitis—a synergistic polymicrobial and opportunistic infection [3,4,5]. According to O’Keefe et al., 2008 [6], nutritional status can directly affect inflammatory pathways and contribute to the onset and progression of chronic diseases. Nutritional issues have been identified as one contributing factor to the etiopathogenesis of periodontal diseases [7,8], as diet influences energy production, genetic modulation, and protein expression. In particular, susceptibility to periodontal diseases may be related to the consumption of lipids [9].

Lipids are essential dietary constituents that occur in a variety of forms: as fats and oils, phospholipids, steroids, and carotenoids [10]. Lipids are essential components of biological membranes, which are largely composed of phospholipids, glycolipids, and sterols. Phospholipids influence the structure and function of the cell membrane [11,12,13]. Among the components of phospholipids are fatty acids, which are classified as saturated, unsaturated, and polyunsaturated based on the number of carbon double bonds (none, one, and two or more, respectively).

Regarding polyunsaturated fatty acids (PUFAs), the human being is not able to produce linoleic acid (omega-6) and α-linolenic acid (omega-3) in the sufficient amounts. In other words, the requirement for these acids is higher than the endogenous supply [14,15] and therefore they must be included in the diet through the ingestion of specific foods or supplements. Once in the body, linoleic acid is converted by means of desaturation and elongation enzymes to arachidonic acid (AA), which is a precursor for many pro-inflammatory mediators. By means of the same enzymes, α-linolenic acid is converted to eicosapentaenoic acid (EPA) followed by conversion into resolvins (E series 1–4) with predominantly anti-inflammatory, vasodilatation and anti-aggregation properties [16,17]. Further desaturation and elongation of EPA results in docosahexaenoic acid (DHA) which is metabolized into D-series of resolvins (Resolvin1–6), protectins and maresins with anti-inflammatory and immunomodulatory properties [18]. As EPA is in competition with AA regarding COX and LOX, the AA:EPA ratio level determines the metabolic status in being more pro-inflammatory/pro-aggregatory or more anti-inflammatory/anti-aggregatory. It has been shown that EPA supplementation results in lower plasma levels of these pro-inflammatory metabolites and in a lower AA:EPA ratio being anti-inflammatory [19].

The ideal dietary intake of omega-6 and omega-3 fatty acids to obtain an optimal AA:EPA ratio is still widely debated in the literature. High dietary intake of omega-6 relative to omega-3 may result in adverse health effects and an increased prevalence of chronic diseases associated with prolonged inflammation [20]. Increased consumption of omega-3 can, therefore, improve the AA:EPA ratio and protect against inflammation. Previous studies have shown that dietary intake of omega-3 appears to reduce the risk of many diseases and disorders, including coronary artery disease [21], type II diabetes [22], rheumatoid arthritis [23], asthma [24], depression [25], and cancer [26].

As mentioned above periodontitis is a synergistic polymicrobial and opportunistic infection. This inflammatory disease is characterized by the formation of deepened periodontal pockets with a pathogenic subgingival microbial biofilm on the root surface and breakdown of the connective tissue between teeth and supporting alveolar bone. Evaluation of the periodontal condition is performed by assessment of the probing pocket depth (PPD) and clinical attachment loss (CAL). A fundamental aspect of the treatment of periodontitis is the reduction of the subgingival microbial biofilm by means of mechanical root instrumentation and removal of the supragingival biofilm by mean of oral hygiene instructions, i.e., non-surgical periodontal therapy (NSPT). This treatment results in PPD reduction and CAL gain. In respect to this treatment, it is interesting that PUFAs have demonstrated also antimicrobial properties [27]. Furthermore, it has been shown that both EPA and DHA reduced significantly bacterial strains in a multi-species subgingival biofilm model including *Porphyromonas gingivalis* and *Aggregatibacter actinomycetemcomitans* [28]. Therefore, it may be not unlikely that in addition to the anti-inflammatory effect, the antimicrobial effect of PUFAs could contribute to the effect of periodontal treatment.

Considering the importance of EPA and DHA in the anti-inflammatory process, supplementation with omega-3 may improve conditions associated with periodontal inflammation. For example, previous studies in animal models found that individuals receiving omega-3 supplements experienced less alveolar bone loss and produced fewer pro-inflammatory cytokines [29,30]. Meanwhile, a cross-sectional study by Naqvi et al. (2010) [31] used data collected between 1999 and 2004 from 9182 adults (≥20 years) as part of the NHANES (National Health and Nutrition Examination Survey) in the United States to demonstrate that dietary intake of DHA was associated with a lower prevalence of periodontitis. Dietary intake of EPA showed a similar, but more modest, relationship. 

From a clinical perspective, the few studies investigating omega-3 supplementation to date exhibit many limitations—having employed different methodological designs, populations and numbers of participants, periodontal diagnoses, treatments, and supplementation protocols—that make it difficult to accurately interpret the results and draw conclusions. Consequently, the findings of systematic reviews analyzing these studies [32,33,34] should also be interpreted with caution. The null hypothesis was that supplementation of PUFAs does not have a positive effect on the results of periodontal treatment. Therefore, the objective of the current study is to examine, through a systematic review, the impact of omega-3 supplementation during NSPT on periodontal clinical parameters.

Therefore, the objective of the current study is to examine, through a systematic review, the impact of omega-3 supplementation of on periodontal clinical parameters.

## 2. Methodology

This systematic review was conducted based on the Cochrane Handbook for Systematic Reviews of Interventions [35] and recommendations developed to strengthen reporting of systematic reviews and meta-analyses (PRISMA) [36]. This review is registered under number CRD42021234746 in PROSPERO, the International Prospective Register of Systematic Reviews.

### 2.1. Search Strategy

This systematic review sought randomized clinical trials (RCTs) that evaluated the effects of omega-3 supplementation, associated with non-surgical treatment of periodontitis, on periodontal clinical parameters. The following PICO question was asked: “What is the possible effect of supplementation with omega-3, in addition to non-surgical periodontal therapy, on clinical periodontal parameters?” From January to July 2021, searches were performed in five databases: Cochrane, Embase, Google Scholar (grey literature), PubMed, and Web of Science. The search strategies used for each database are shown in Table 1. Studies meeting the following criteria were retained for further evaluation:Population: patients >18 years old with a diagnosis of periodontitisIntervention: non-surgical periodontal treatment with oral intake of omega-3 as the sole supplementComparison: non-surgical periodontal treatment with or without a control (i.e., use of a placebo)Main outcome: periodontal probing depth (PPD)

### 2.2. Screening and Selection

Titles and abstracts returned through each database search were independently evaluated by two reviewers (LMM and FBP) to select those meeting the inclusion criteria. Studies that did not provide enough information in the title or abstract to determine their eligibility for inclusion were thoroughly reviewed. Studies were categorized as eligible, ineligible, or questionable. Disagreements were resolved by consensus; in those cases where conflict persisted, the final decision was made by a third reviewer (SCG). Articles that met the inclusion criteria were read in full and processed for data extraction. 

The selection of study titles and abstracts was performed according to the following inclusion criteria:Randomized clinical trialsStudies of non-surgical treatment of periodontitis with omega-3 supplementation, with or without a control treatmentA minimum of 3 months of experimental period and supplementation lengthNo language restriction

Exclusion criteria were:Study length of <3 monthsAssociation of omega-3 with other forms of supplementation or medicationAbsence of periodontal treatmentAbsence of supplementation with omega-3Pilot studiesRestricted to evaluation of gingivitisUse of ineligible study designUse of surgical periodontal therapy

### 2.3. Risk of Bias Assessment

The risk of bias for each study was assessed by two independent reviewers (LMM and FBP) using the Cochrane risk of bias tool version 2 (RoB 2). The resulting score is based on five domains and classifies studies as having low risk, some concerns, or high risk of bias. The five evaluated criteria were the randomization process, deviations from intended interventions, missing outcome data, outcome measurement, and selection of reported outcomes. The overall risk of bias depended on the ranking of each domain (Figure 1).

### 2.4. Data Extraction

Data extraction was performed independently by two reviewers. In cases of discordance, a third reviewer would be decisive (SCG). The following data were extracted from each study: main authorship, study title, country, type of study, study period, supplementation period length, number of participants, age, systemic conditions, periodontal diagnosis, placebo use, dosage of omega-3 supplementation, capsules intake control, oral hygiene instructions, adverse effects, author’s conflict of interest statement, and initial and final values for each periodontal clinical parameter.

### 2.5. Data Analysis

Relevant data and comparisons of the included studies were presented descriptively. When applicable, delta values (i.e., the difference between the baseline and final values) were calculated for the clinical parameters. Due to the high methodological heterogeneity of the studies, it was decided to do not perform meta-analysis. 

### 2.6. Classification of Evidence

The GRADE system (Grading of Recommendations Assessment, Development, and Evaluation) was used to classify the evidence produced through this review [37]. Two reviewers assessed the quality of evidence and the strength of the recommendations (LMM and SCG) according to the following factors: risk of bias, consistency of results, objectivity of evidence, accuracy of data, publication bias, and magnitude of effect. 

## 3. Results

The initial database search resulted in 1556 articles, of which 193 were excluded due to duplication and 1345 were excluded after reading the titles and abstracts. The following criteria led to exclusion: Absence of periodontal treatment (n = 4)Absence of supplementation with omega-3 (n = 5)Association of omega-3 with other forms of supplementation or medication (n = 28)Use of ineligible study design (n = 56)Local application of omega-3 (n = 1)Restricted evaluation of gingivitis (n = 1)Lack of connection with periodontal evaluation (n = 1250)

Screening of titles and abstracts by both independent reviewers (LMM and FBP) resulted in retention of the same 28 articles (Figure 2). After reading all the articles thoroughly, additional exclusions were made due to the following criteria: Study length of <3 months (n = 1)Association of omega-3 with other forms of supplementation or medication (n = 2)Absence of periodontal treatment (n = 2)Absence of supplementation with omega-3 (n = 6)Pilot studies (n = 2)Restricted to evaluation of gingivitis (n = 1)Use of ineligible study design (n = 5)Use of surgical periodontal therapy (n = 1)

Ultimately, eight studies were included in this systematic review. The general characteristics of each study are presented in detail in Table 2. The number of participants in the studies ranged from 30 to 60, and the study length from 3 to 6 months. All studies included adult participants, of at least 22 years. In the same table, it can be noted that one study included people with diabetes (type 2 diabetes mellitus) [38], other obese individuals [39], and a third menopausal participants [40].

The diagnosis of the patients in the included studies varied from moderate/severe chronic [41], untreated advanced chronic [43], chronic [38,42], generalized chronic [39,40] to stage II/III periodontitis [44] and stage III/IV periodontitis [45] according to the most recent classification [46]. Six out of eight studies presented detailed criteria for patient selection. Four studies used presence of sites with PPD (probing depth) ≥ 5 mm [38,40,41,42] and two studies presence of sites with PPD ≥ 6 mm [43,45]. Of the remaining two studies one study included patients with stage II and III grade B periodontitis [44] and the other mentioned only chronic periodontitis, both without any further information [39]. 

The number of teeth, as far as presented, varied greatly among studies. Two studies reported the number of teeth examined [42,43]. Four authors did not report the number of teeth examined but defined the inclusion criteria based on the minimum number of teeth present: at least 6 [40], 15 [38], ≥18 [45] and 20 [39]. In contrast, Deore et al. (2014) [41] and Shalaby and Morsy (2019) [44] did not provide any information about number of teeth. Besides, information about the number of sites in which the clinical parameters were assessed varied as well. For example, four studies evaluated clinical parameters at six sites per tooth [39,40,42,44,45], one at four sites [41] and the remaining two studies did not provide any information about the number of sites examined [38,43]. 

PPD and CAL were evaluated in all studies, and presence of plaque in seven of them [39,40,41,42,43,44,45] (Appendix A). Gingival inflammation was assessed in seven studies by means of the Gingival Index (GI) [47] of which in one study [43] the modified GI (MGI) [48] was used and in another one [44] the GI inflammation index (GI II) [49]. In two studies both the Sulcus Bleeding Index and the GI were applied [39,41] and in two other studies [42,43] a combination of the GI and BOP (bleeding assessment after PPD) was used. In terms of results, there were no differences which method was performed. One study [45] used BOP assessment as sole parameter of inflammation. 

The length of the omega-3-supplementation period coincided with the experimental period in all studies, starting at baseline. However, the dose of omega-3 administered per day in the studies varied widely, from 50.88 mg to 3000 mg. Five studies reported the exact amount of EPA and DHA used [39,40,41,42,45]. Capsule intake control was reported in seven studies. [39,40,41,42,43,44,45]. Mild adverse effects (nausea or fish breath) were reported in one study in 6 out of 30 participants [45], while all other studies reported no side effects (Table 2). No authors declared conflict of interest regarding their participation.

Randomization of participants was in all studies but one [42] performed before treatment. The provided periodontal therapy included in six studies scaling and root planning (SRP) in conjunction with oral hygiene instructions (OHI) and in two studies SRP solely [38,39]. Repetition of OHI during the experiment was carried out in the sixth and twelfth weeks of the study by Deore et al. (2014) [41], in the third and sixth month by Elgendy and Kazem (2018) [40] and Shalaby and Morsy (2019) [44], and in the third week after baseline by Stando et al. (2020) [45]. At baseline no differences were observed between test and control groups regarding PPD, CAL and gingival inflammation assessments. This was found for plaque as well in seven studies. Only in one study baseline plaque scores were in the control group higher compared to that of the test group (Table 3) [45]. With regard to plaque reduction during the study, results of all studies showed no differences between test and control groups (Appendix A). 

Comparison of the treatment results of the included studies is presented in Table 3. In all studies it was found that the provided treatment resulted both in test- and control groups in significant PPD reduction, CAL gain and less inflammation. Omega-3 supplementation resulted in four of eight studies in significant greater PPD reduction and CAL gain compared to the control group [40,41,43,44]. In one study [45] a significant CAL gain in favor of the test group did not coincide with greater PPD reduction. A significant reduction of gingival inflammation in favor of the test group was found in two studies [40,41]. There was no indication that the amount omega-3 supplementation was related to a possible treatment effect.

The estimated risk of bias of the included studies is shown in detail in Figure 2. Selection bias was observed in two studies [38,43] where insufficient information was provided regarding the confidentiality of the allocation of experimental groups. The blinding of participants was not possible in those studies that did not use a placebo in their interventions [39,43,44,45]. The lack of information about which teeth were examined, led to the classification of one study, in domain two, as high risk [38]. The absence of a placebo in four studies [39,43,44,45] and led to their classification, in domain two, as moderate risk. The overall assessment of the potential risk of bias resulted in six studies with a high risk of bias and only two with low risk. Due to the serious risk of bias and inconsistency found here, the studies included have moderate certainty regarding PPD, CAL and gingival inflammation results. Table 4 summarizes the factors used to assess the quality of evidence according to the GRADE system. 

## 4. Discussion

This systematic review investigated the effect of omega-3 supplementation on NSPT in randomized clinical trials. In this respect, the omega-3 dose administered and the duration time of the administration are important variables. It has been shown that incorporation rates of EPA and DHA vary according to the sample evaluated; incorporation may take days (for triglycerides and phosphatidylcholine in plasma fractions), months (6–9 months for mononuclear cells), or years (for adipose tissue) [50]. The benefits of supplementation also manifest over varying time intervals for different diseases. In addition, the greatest benefits with omega-3 are believed to come from the daily intake of fatty fish (salmon, tuna, mackerel, herring and sardines) and from some seeds and vegetables [51]. However, none of the studies included here evaluated the current dietary habits of the participants, which could lead to significant bias. Besides, the most current international recommendations for intake of DHA and EPA are based mainly on epidemiological and clinical studies that aimed to assess the benefits of omega-3 intake for the treatment of cardiovascular diseases, suggesting a dose ranging from 0.4 to 1.0 g/day [52,53,54]. 

In the studies included here, the highest omega-3 dosage used was 4400 mg per day consisting of 2600 mg of EPA and 1800 mg of DHA [45]; compared to the control group this resulted in more CAL gain but not more PPD and BOP reduction. These results might be partially explained by the short supplementation time (three months), which contradicts data suggesting a minimum period of six months for EPA and DHA to be incorporated into mononuclear cells [50]. The three-months supplementation period was used in five of the eight included studies. In two studies a significant effect on PPD reduction in favor of the test group was found and in three not. The two positive studies used a daily dosage of 300 mg omega-3 [41] and 2000 mg [43], respectively. The daily omega-3 dosage of the three negative studies was 1000 mg in two studies [38,39] and 4400 mg in the study as mentioned above [45]. With regard to CAL, the three months supplementation resulted in three studies with a significant CAL gain [41,43,45] and in two not [38,39]. These findings may suggest that three months omega-3 supplementation has a greater effect on CAL gain than PPD reduction. However, to put it in perspective, the PPD reduction in control groups of the above five studies varied from 0.94 mm to 2.0 mm (mean 1.3 mm) which is within the expected range of the effect of non-surgical periodontal therapy (NSPT) after three months [55]. CAL gain in the control group varied from 0.8 mm to 2.0 with a mean value of 1.3 mm, comparable to the PPD reduction. This is in contrast to the results as presented by Cobb (2002) [55] showing CAL gain after NSPT of about 0.5 mm. This may indicate bias in the supplementation studies. 

Regarding the performed periodontal therapy, two three-month 1000 mg omega-3 supplementation studies without significant differences between test and control [38,39] did not carry out oral hygiene instructions, in contrast to the other studies. The lack of oral hygiene instructions is an important issue because it has been shown that supragingival plaque control reduces PPD values, improves CAL gain [56,57], as well as maintenance the results obtained after NSPT [58]. On the other hand, these two studies were the only studies that included participants with systemic conditions that interfere with the individual’s inflammatory state and are considered a risk factor (diabetes mellitus) or risk indicator (obesity) for periodontal diseases. Including participants with systemic conditions that alter the host response to periodontal treatment may be premature, as existing evidence is insufficient to identify the patient profile and severity of periodontal disease that will benefit from omega-3 supplementation in individuals with such conditions. 

A longer period of daily omega-3 supplementation for 6 months was used in three studies. In two studies a significant effect was found for PPD reduction and CAL gain in favor of the test group [40,44] and in one study not [42]. This discrepancy may be linked to the omega-3 dosage used. Elgendy and Kazem (2018) [40] used 1000 mg omega-3 and Shalaby and Morsy (2019) [44] 3000 mg omega-3. In contrast, Keskiner et al. (2017) [42] used 50.88 mg omega-3 which may be too low to be effective. These two positive 6-month studies would suggest that supplementation of NSPT with daily omega-3 ≥ 1000 mg over a longer period of time is effective, confirming the time period needed for substantial incorporation of EPA and DHA in mononuclear cells [50]. Nevertheless, as in the three-month studies, the amount of PPD reduction in the control groups is within the expected range of the effect of non-surgical periodontal therapy (NSPT) after six months [55]. However, regarding CAL gain this was not the case. Both Elgendy and Kazem (2018) [40] and Shalaby and Morsy (2019) [44] show CAL gain of 1.5 mm, which is according to the data provided by Cobb (2002) [55] highly exceptional. 

The above observations underscore that new RCTs on periodontal treatment and omega-3 supplementation should include, not only six months of omega-3 supplementation, but also evaluation of the gingival condition in combination with the level of supragingival plaque control on a regular basis as well as evaluation of the regular consumption of omega-3 from the daily diet. In addition, the medical condition of study participants should be checked in order to selects patients suffering from no other disease or condition than periodontal disease. 

The studies included in this systematic review present several limitations. The few RCTs available to date exhibit a high degree of methodological heterogeneity, which precluded the performance of a meta-analysis. Based on the RoB tool, six studies included RCTs were also characterized by a high risk of bias. Meanwhile, the GRADE system classified the quality of evidence for the results of this review as ‘moderate.’ These limitations prevent us from reaching a satisfactory conclusion on the benefits of omega-3 supplementation. 

## 5. Conclusions

Notwithstanding all limitations, a promising effect of omega-3 supplementation on the effect of NSPT was present in two six months studies with daily omega-3 supplementation ≥ 1000 mg. Both showing significant more PPD reduction and CAL gain after NSPT in the test group compared to the control. These findings encourage performing RCTs with better-defined treatment protocols and greater methodological rigor. 

## Figures and Tables

**Figure 1 nutrients-14-01838-f001:**
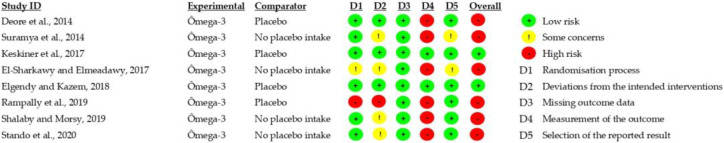
Risk of bias assessment of included studies.

**Figure 2 nutrients-14-01838-f002:**
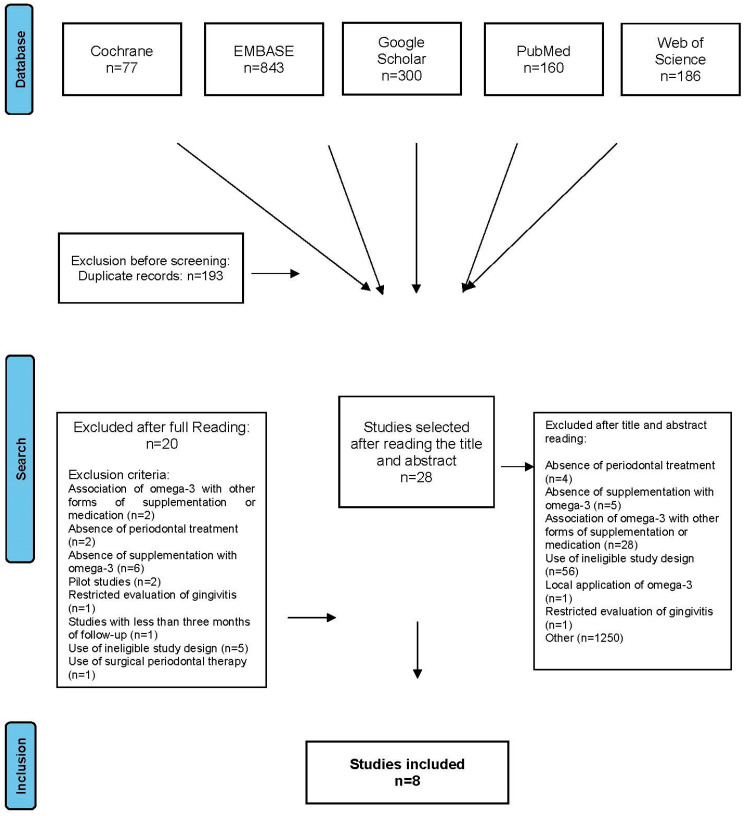
Flowchart of the process used to select studies for inclusion in the systematic review.

**Table 1 nutrients-14-01838-t001:** Search strategies used to identify studies for inclusion in the systematic review, customized for each database.

Database	Strategy
Cochrane	(Periodontitis OR Chronic periodontitis OR Periodontal disease OR probing pocket depth OR periodontal pocket) AND (Intervention OR Therapy OR Treatment OR Scaling and root planning OR SRP OR nonsurgical periodontal therapy OR non-surgical therapy OR Periodontal treatment OR Periodontal therapy) AND (fatty acids, omega-3 OR docosahexaenoic acids OR eicosapentaenoic acids OR fatty acids OR fish oils OR omega-3 OR ω-3 OR n-3 OR PUFA OR Long chain fatty acids)
Embase	#1 ‘fatty acids’ OR ‘omega-3’ OR ‘docosahexaenoic acids’ OR ‘eicosapentaenoic acids’ OR ‘fish oils’ OR ‘ω-3’ OR ‘n-3’ OR ‘pufa’ OR ‘long chain fatty acids’ #2 periodontitis OR ‘chronic periodontitis’ OR ‘periodontal disease’ OR ‘probing pocket depth’ OR ‘periodontal pocket’ #3 Intervention OR Therapy OR Treatment #4 ‘scaling and root planing’ OR ‘srp’ OR ‘nonsurgical periodontal therapy’ OR ‘non-surgical therapy’ OR ‘periodontal treatment’ OR ‘periodontal therapy’ #5 (#3 OR #4) #6 (#1 AND #2 AND #5)
Google Scholar	Periodontal diseases and non-surgical therapy and omega-3
PubMed	#1 (fatty acids, ômega-3 [MeSH Terms]) OR (docosahexaenoic acids [Text Word]) OR (eicosapentaenoic acids [Text Word]) OR (fatty acids [Text Word]) OR (fish oils [Text Word]) OR (ômega-3 [Text Word]) OR (ω- 3 [Text Word]) OR (n-3 [Text Word]) OR (PUFA [Text Word]) OR (Long chain fatty acids [ Text Word]) #2 (periodontal diseases [ MeSH Terms]) OR (periodontitis [MeSH Terms]) OR (Chronic periodontitis [MeSH Terms]) OR (probing pocket depth [Text Word]) OR (periodontal pocket [Text Word]) #3 (Scaling and root planing [Text Word]) OR (SRP [Text Word]) OR (nonsurgical periodontal therapy [Text Word]) OR (non-surgical therapy [Text Word]) OR (Periodontal treatment [Text Word]) OR (Periodontal therapy [Text Word]) #4 (Therapy [MeSH Terms]) OR (Treatment [Text Word]) OR (Intervention [Text Word]) #5 #3 OR #4 #6 (#1) AND (#2)) AND (#5)
Web of Science	#1 (TS = (Fatty acids OR omega-3 OR docosahexaenoic acids OR eicosapentaenoic acids OR Fish oil OR ω-3 OR n-3 OR PUFA OR Long chain fatty acids)) #2 (TS = (Periodontitis OR Chronic periodontitis OR Periodontal disease OR probing pocket depth OR periodontal pocket)) #3 (TS = (Intervention OR Therapy OR Treatment)) #4 (TS = (Scaling and root planing OR SRP OR nonsurgical periodontal therapy OR non-surgical therapy OR Periodontal treatment OR Periodontal therapy)) #5 (#3 OR #4) #6 (#1 AND #2 AND #5)

**Table 2 nutrients-14-01838-t002:** Study characteristics of selected studies.

Author/Year/ (Reference Number)	Sample Size	Age Years	Systemic Conditions	Periodontitis Diagnosis	Number of Teeth Present (and Mean Number of Teeth Examined)	Study Duration	Omega-3		Capsules Intake Control	Side Effects Reported	Conflict of Interest
Country	(n/Group)				Number of Sites Examined		EPA Dose per Day	DHA Dose per Day			
Study Period					Periodontal Parameters Evaluated						
Deore et al., 2014 [41]	60	T: 45.40 ± 40.90	Healthy	Moderate/severe chronic periodontitis	NR (NR)		180 mg			None	Not declared
India	(T:30/C:30)				4 sites	3 months		120 mg	Yes		
3 months		C: 44.47 ± 5.20			Plaque, GI, SBI, PPD, CAL						
Suramya et al., 2014 [39]	40	T + C > 30	Obese	Generalized chronic periodontitis	At least 20 (NR)		550 mg			None	Not declared
India	(T:20/C:20)				6 sites	3 months		450 mg	Yes		
3 months					Plaque, GI, SBI, PPD, BOP, CAL						
Keskiner et al., 2017 [42]	30	T: 40.87 ± 9.7	Healthy	Chronic periodontitis	NR (T: 25.87 ± 1.19 C: 26.33 ± 1.23)		12.5 mg			None	Not declared
Turkey	(T:15/C:15)	C: 42.54 ± 5.82			6 sites	6 months		38.38 mg	Yes		
3 months					Plaque, GI, PPD, BOP, CAL						
El-Sharkawy and Elmeadawy, 2017 [43]	34	T: 45.75 ± 2.05	Healthy	Untreated advanced chronic periodontitis	At least 18 (T: 24.1 ± 3.2 C: 25.1 ± 2.1)		2000 mg Omega-3			None	Not declared
Egypt	(T:17/C:17)	C: 47.82 ± 2.21			NI	3 months			Yes		
3 months					Plaque, MGI, PPD, BOP, CAL						
Elgendy and Kazem, 2018 [40]	50	T: 50.24 ± 3.04	Post-menopause	Generalized chronic periodontitis	At least 6 (NR)		600 mg			None	Not declared
Egypt	(T:25/C:25)	C: 51.44 ± 3.36			6 sites	6 months		400 mg	Yes		
6 months					Plaque, GI, PPD, CAL						
Rampally et al., 2019 [38]	42	T + C 30–65	Diabetes II	Chronic periodontitis	At least 15 (NR)		1000 mg			None	Not declared
India	(T:14)				NR	3 months			No		
3 months	(C:14)				GI, PPD, CAL						
Shalaby and Morsy, 2019 [44]	45	T + C 35–55	Healthy	Stage II and III, grade B periodontitis	NR (NR)		3000 mg Omega-3			None	Not declared
Egypt	(T1:15) (T2:15)				6 sites	6 months			Yes		
6 months	(C:15)				Plaque, GI II, PPD, CAL						
Stando et al., 2020 [45]	40	T: 45 ± 8	Healthy	Stage III and IV periodontitis	At least 18 (NR)					Nausea and irritating fish- scented halitosis (6 subjects)	Not declared
Poland	(T:16/C:14)	C: 54 ± 11			6 sites	3 months	2600 mg	1800 mg	Yes		
3 months					Plaque, PPD, BOP, CAL						

GI: gingival index; MGI: modified gingival index; GI II: GI inflammation index; SBI: sulcus bleeding index; PPD: probing pocket depth; BOP: bleeding on probing; CAL: clinical attachment loss; NR: not reported.

**Table 3 nutrients-14-01838-t003:** Mean values (standard deviation) of clinical periodontal parameters of test and control groups at baseline and final examination.

Authors, Year, [Reference Number]	Groups	PPD (mm)		*p*-Value	Delta	CAL		*p*-Value	Delta	GI		*p*-Value	Delta
		Baseline	Final	Inter-Group	PPD	Baseline	Final	Inter- Group	CAL	Baseline	Final	Inter-Group	GI
Deore et al., 2014 [41]	T	4.26 ± 1.10	2.15 ± 0.53	*p* < 0.05	2.11	5.53 ± 0.95	2.73 ± 0.98	*p* < 0.05	2.80	1.93 ± 0.29	1.12 ± 0.14	*p* < 0.01	0.81
	C	4.05 ± 1.03	2.77 ± 0.47		1.28	5.20 ± 0.90	3.72 ± 0.62		1.48	2.04 ± 0.34	1.43 ± 0.33		0.61
Suramya et al., 2014 [39]	T	5.45 ± 0.42	4.30 ± 0.79	*p* > 0.05	1.15	5.55 ± 0.52	4.56 ± 0.80	*p* > 0.05	0.99	2.38 ± 0.31	1.08 ± 0.0.16	*p* > 0.05	1.30
	C	5.50 ± 0.53	4.56 ± 0.80		0.94	5.42 ± 0.37	4.56 ± 0.80		0.86	2.31 ± 0.31	1.05 ± 0.12		1.26
Keskiner et al., 2017 [42]	T	3.72 (2.23–4.75) ^4^	2.46 (1.83–3.32)	*p* > 0.05	1.26	4.59 (3.04–5.31) ^4^	3.53 (2.42–4.08)	*p* > 0.05	1.06	1.82(1.49–2.12) ^4^	1.23(0.67–1.44)	*p* > 0.05	0.59
	C	3.73 (2.43–4.25)	2.38 (2.04–3.23)		1.35	4.20 (2.73–5.33)	3.10 (2.68–4.16)		1.10	1.68(1.45–1.92)	1.18(1.06–1.34)		0.50
El-Sharkawy and Elmeadawy, 2017 [43]	T	4.76 ± 0.84	2.24 ± 0.27	*p* < 0.001	2.52	5.16 ± 0.52	2.78 ± 0.38	*p* < 0.001	2.38	2.32 ± 0.19 ^1^	0.66 ± 0.16	*p* > 0.05	1.66
	C	4.46 ± 0.57	3.37 ± 0.64		1.09	5.08 ± 0.46	3.84 ± 0.49		1.24	2.27 ± 0.13	0.72 ± 0.14		1.55
Elgendy and Kazem, 2018 [40]	T	6.00 ± 0.59	3.46 ± 0.49	*p* < 0.05	2.54	5.96 ± 0.61	3.40 ± 0.50	*p* < 0.05	2.56	1.98 ± 0.30	0.30 ± 0.23	*p* < 0.05	1.68
	C	5.84 ± 0.61	4.29 ± 0.75		1.55	5.79 ± 0.72	4.06 ± 0.59		1.73	2.06 ± 0.39	0.55 ± 0.32		1.51
Rampally et al., 2019 [38]	T	6.71 ± 0.47	4.71 ± 0.47	*p* > 0.05	2.00	5.71 ± 0.47	3.71 ± 0.47	*p* > 0.05	2.00	2.03 ± 0.30	1.26 ± 0.44	*p* > 0.05	0.77
	C	6.43 ± 0.51	4.43 ± 0.51		2.00	5.43 ± 0.51	3.43 ± 0.51		2.00	1.96 ± 0.44	1.14 ± 0.57		0.82
Shalaby and Morsy, 2019 [44]	T	5.47 ± 0.94	3.14 ± 0.64	*p* < 0.01	2.33	4.08 ± 0.96	2.60 ± 0.52	*p* < 0.01	1.48	1.95 ± 0.53 ^2^	0.47 ± 0.25	*p* > 0.05	1.48
	C	5.82 ± 0.61	4.31 ± 0.84		1.51	4.28 ± 0.57	3.49 ± 0.85		0.79	2.03 ± 0.49	0.57 ± 0.30		1.46
Stando et al., 2020 [45]	T	5.0 ± 0.5	3.7 ± 0.7	*p* > 0.05	1.3	5.8 ± 0.8	4.4 ± 1.1	*p* < 0.05	1.4	28 ± 16 ^3^	14 ± 6	*p* > 0.05	14
	C	5.1 ± 0.8	4.0 ± 0.7		1.1	6.1 ± 1.1	5.3 ± 1.0		0.8	36 ± 19	21 ± 7		15

T: test group; C: control group; PPD: probing pocket depth; CAL: clinical attachment loss; GI: gingival index; **^1^** modified gingival index; **^2^** GI modification; ^**3**^ bleeding on probing (%); **^4^** median and percentiles (25–75).

**Table 4 nutrients-14-01838-t004:** Summary of findings based on quality and body of evidence. GRADE.

Participants (Studies)	Risk of Bias	Inconsistency	Indirectness	Imprecision	Other Considerations	Overall Certainty of Evidence	Impact
Probing depth (PPD)							
302 (8 RCTs)	very serious	serious	not serious	not serious	All plausible residual confounding would reduce the demonstrated effect dose response gradient	+++○	Six studies have a high risk of bias. In general, studies report different results for PPD
Clinical attachment loss (CAL)							
302 (8 RCTs)	very serious	serious	not serious	not serious	All plausible residual confounding would reduce the demonstrated effect dose response gradient	+++○	Six studies have a high risk of bias. In general, studies report different results for CAL
Gingival inflammation							
302 (8 RCTs)	very serious	serious	not serious	not serious	All plausible residual confounding would reduce the demonstrated effect dose response gradient	+++○	Six studies have a high risk of bias. In general, studies report different results for gingival inflammation

+++○, Moderate.

## Data Availability

Data can be obtained from the referred publications.

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
