# Peer review of "The Impact of Omega-3 Supplements on Non-Surgical Periodontal Therapy: A Systematic Review"

_nutrients, 2022, doi:10.3390/nu14091838_

Round 1

Reviewer 1 Report

Miller et al., reviewed 8 peer-reviewed articles on omega-3 as a periodontal therapy. Despite the small number of studies included and many contracting results, the manuscript is well-written overall with a clear aim, detailed descriptions of each study, and limitations 
This manuscript can be improved:
1. Table 2 and 3: include reference number 
2. Conclusion is the exact same sentence as the last part of the abstract. It needs to be revised and expanded.
3. Author connects omega-3 to immune response marker but there is no mention of the microbiome and its metabolites. It would be interesting to see the connection between omega-3 and the oral microbiome.

Author Response

Dear reviewer,

Thanks for your time and suggestions that let us improve the manuscript. I hope we could answer your queries accordingly. All the changes provided are tracked along with the text.

This manuscript can be improved:

  1. Table 2 and 3: include reference number:
    1. The references numbers were added and are tracked in yellow.
  1. Conclusion is the exact same sentence as the last part of the abstract. It needs to be revised and expanded.
    1. The conclusion now reads: Notwithstanding all limitations, a promising effect of omega-3 supplementation on the effect of NSPT was present in two six months studies with daily omega-3 supplementation ≥ 1000 mg. Both showing significant more PPD reduction and CAL gain after NSPT in the test group compared to the control. These findings encourage performing RCTs with better-defined treatment protocols and greater methodological rigor.
  1. Author connects omega-3 to immune response marker but there is no mention of the microbiome and its metabolites. It would be interesting to see the connection between omega-3 and the oral microbiome.
    1. Additional information was provided as reads: A fundamental aspect of the treatment of periodontitis is the reduction of the subgingival microbial biofilm by means of mechanical root instrumentation and removal of the supragingival biofilm by mean of oral hygiene instructions i.e. non-surgical periodontal therapy (NSPT). In this respect it is interesting that PUFAs have demonstrated also antimicrobial properties [27]. Furthermore, it has been shown that both EPA and DHA reduced significantly bacterial strains in a multi-species subgingival biofilm model including Porphyromonas gingivalis and Aggregatibacter actinomycetemcomitans [28]. Therefore, it may be not unlikely that in addition to the anti-inflammatory effect, the antimicrobial effect of PUFAs could contribute to the effect of periodontal treatment.

Reviewer 2 Report

Dear authors,

Thank you for submitting your valuable work to the journal. The topic of the review is interesting, focusing on relavant existing information on the use of Omega-3 as an adjunvat in non-surgical periodontal therapy. The purpose of non-surgical therapy, alonside the elimination of subgingival bacterial plaque, is to reduce inflammation and to trigger local healling process of the jonctional epithelium. Thus, the use of adjuvants of this process could be of interest.

There are some comments that I would make in order to improve the paper's accuracy and maximize its scientific impact:

  • in Abstract please explain RCT acronym
  • in the Introduction please contract information on various types of PUFAs
  • in the Introduction please add information on the principles of non-surgical periodontal therapy
  • Please add a nulle Hypothesis for the review
  • Please move Exclusion Criteria for studies in the Materials and Methods section
  • Please rephrase Conclusion for more precise and easier understanding

We look forward to receiving the revised version of your manuscript!

Kind regards!

Author Response

Dear reviewer,

Thanks for your time and suggestions that let us improve the manuscript. I hope we could answer your queries accordingly. All the changes provided are tracked along with the text.

  1. Abstract please explain RCT acronym
    1. Randomized clinical trials (RCT)
  1. in the Introduction, please contract information on various types of PUFAs
    1. Regarding polyunsaturated fatty acids (PUFAs), the human being is not able to produce linoleic acid (omega-6) and α-linolenic acid (omega-3) in the sufficient amounts. In other words, the requirement for these acids is higher than the endogenous supply [14, 15] and therefore they must be included in the diet through the ingestion of specific foods or supplements. Once in the body, linoleic acid is converted by means of desaturation and elongation enzymes to arachidonic acid (AA), which is a precursor for many pro-inflammatory mediators. By means of the same enzymes, α-linolenic acid is converted to eicosapentaenoic acid (EPA) followed by conversion into resolvins (E series 1-4) with predominantly anti-inflammatory, vasodilatation and anti-aggregation properties [16, 17]. Further desaturation and elongation of EPA results in docosahexaenoic acid (DHA) which is metabolized into D-series of resolvins (Resolvin1-6), protectins and maresins with anti-inflammatory and immunomodulatory properties [18].
  1. in the Introduction, please add information on the principles of non-surgical periodontal therapy.
    1. As mentioned above periodontitis is a synergistic polymicrobial and opportunistic infection. This inflammatory disease is characterized by the formation of deepened periodontal pockets with a pathogenic subgingival microbial biofilm on the root surface and breakdown of the connective tissue between teeth and supporting alveolar bone. Evaluation of the periodontal condition is performed by assessment of the probing pocket depth (PPD) and clinical attachment loss (CAL). A fundamental aspect of the treatment of periodontitis is the reduction of the subgingival microbial biofilm by means of mechanical root instrumentation and removal of the supragingival biofilm by mean of oral hygiene instructions i.e. non-surgical periodontal therapy (NSPT). This treatment results in PPD reduction and CAL gain. In respect to this treatment, it is interesting that PUFAs have demonstrated also antimicrobial properties [27]. Furthermore, it has been shown that both EPA and DHA reduced significantly bacterial strains in a multi-species subgingival biofilm model including Porphyromonas gingivalis and Aggregatibacter actinomycetemcomitans [28]. Therefore, it may be not unlikely that in addition to the anti-inflammatory effect, the antimicrobial effect of PUFAs could contribute to the effect of periodontal treatment.
  1. Please add a nulle Hypothesis for the review
    1. The null hypothesis was that supplementation of PUFAs does not have a positive effect on the results of periodontal treatment.
  1. Please move Exclusion Criteria for studies in the Materials and Methods section

Moved: Exclusion criteria were:

  • Study length of < 3 months
  • Association of omega-3 with other forms of supplementation or medication
  • Absence of periodontal treatment
  • Absence of supplementation with omega-3
  • Pilot studies
  • Restricted to evaluation of gingivitis
  • Use of ineligible study design
  • Use of surgical periodontal therapy
  1. Please rephrase Conclusion for more precise and easier understanding
    1. Notwithstanding all limitations, a promising effect of omega-3 supplementation on the effect of NSPT was present in two six months studies with daily omega-3 supplementation ≥ 1000 mg. Both showing significant more PPD reduction and CAL gain after NSPT in the test group compared to the control. These findings encourage performing RCTs with better-defined treatment protocols and greater methodological rigor.

Besides, we changed parts of the discussion to better address important information regarding omega 3 intake (the value of evaluation the current diet) and separated the studies on omega 3, according to the length of supplementation/experimental interventions.

In addition, the greatest benefits with omega-3 are believed to come from the daily intake of fatty fish (salmon, tuna, mackerel, herring and sardines) and from some seeds and vegetables [51]. However, none of the studies included here evaluated the current dietary habits of the participants, which could lead to significant bias. Besides, the most current international recommendations for intake of DHA and EPA are based mainly on epidemiological and clinical studies that aimed to assess the benefits of omega-3 intake for the treatment of cardiovascular diseases, suggesting a dose ranging from 0.4 to 1.0 g/day [52-54].

In the studies included here, the highest omega-3 dosage used was 4400 mg per day consisting of 2600 mg of EPA and 1800 mg of DHA [45]; compared to the control group this resulted in more CAL gain but not more PPD and BOP reduction. These results might be partially explained by the short supplementation time (three months), which contradicts data suggesting a minimum period of six months for EPA and DHA to be incorporated into mononuclear cells [50]. The three-months supplementation period was used in five of the eight included studies. In two studies a significant effect on PPD reduction in favor of the test group was found and in three not. The two positive studies used a daily dosage of 300 mg omega-3 [41] and 2000 mg [42] respectively. The daily omega-3 dosage of the three negative studies was 1000 mg in two studies [38, 39] and 4400 mg in the study as mentioned above [45].  With regard to CAL, the three months supplementation resulted in three studies with a significant CAL gain [41, 42, 45] and in two not [38, 39]. These findings may suggest that three months omega-3 supplementation has a greater effect on CAL gain than PPD reduction. However, to put it in perspective, the PPD reduction in control groups of the above five studies varied from 0.94 mm to 2.0 mm (mean 1.3 mm) which is within the expected range of the effect of non-surgical periodontal therapy (NSPT) after three months [55]. CAL gain in the control group varied from 0.8 mm to 2.0 with a mean value of 1.3 mm, comparable to the PPD reduction. This is in contrast to the results as presented by Cobb (2002) [55] showing CAL gain after NSPT of about 0.5 mm. This may indicate bias in the supplementation studies.

Regarding the performed periodontal therapy, two three-months 1000 mg omega-3 supplementation studies without significant differences between test and control [38, 39] did not carry out oral hygiene instructions, in contrast to the other studies. The lack of oral hygiene instructions is an important issue because it has been shown that supragingival plaque control reduces PPD values, improves CAL gain [56, 57] as well as maintenance the results obtained after NSPT [58]. On the other hand, these two studies were the only studies that included participants with systemic conditions that interfere with the individual's inflammatory state and are considered a risk factor (diabetes mellitus) or risk indicator (obesity) for periodontal diseases. Including participants with systemic conditions that alter the host response to periodontal treatment may be premature, as existing evidence is insufficient to identify the patient profile and severity of periodontal disease that will benefit from omega-3 supplementation in individuals with such conditions.
